# AST-GIN: Attribute-Augmented Spatiotemporal Graph Informer Network for Electric Vehicle Charging Station Availability Forecasting

**DOI:** 10.3390/s23041975

**Published:** 2023-02-10

**Authors:** Ruikang Luo, Yaofeng Song, Liping Huang, Yicheng Zhang, Rong Su

**Affiliations:** School of Electrical and Electronic Engineering, Nanyang Technological University, Singapore 639798, Singapore

**Keywords:** deep learning, attribute-augmented, prediction, weather, spatiotemporal

## Abstract

Electric Vehicle (EV) charging demand and charging station availability forecasting is one of the challenges in the intelligent transportation system. With accurate EV station availability prediction, suitable charging behaviors can be scheduled in advance to relieve range anxiety. Many existing deep learning methods have been proposed to address this issue; however, due to the complex road network structure and complex external factors, such as points of interest (POIs) and weather effects, many commonly used algorithms can only extract the historical usage information and do not consider the comprehensive influence of external factors. To enhance the prediction accuracy and interpretability, the Attribute-Augmented Spatiotemporal Graph Informer (AST-GIN) structure is proposed in this study by combining the Graph Convolutional Network (GCN) layer and the Informer layer to extract both the external and internal spatiotemporal dependence of relevant transportation data. The external factors are modeled as dynamic attributes by the attributeaugmented encoder for training. The AST-GIN model was tested on the data collected in Dundee City, and the experimental results showed the effectiveness of our model considering external factors’ influence on various horizon settings compared with other baselines.

## 1. Introduction

Traffic information forecasting plays an important role in smart city management. Generally speaking, traffic information contains the link speed, traffic flow, vehicle density, traveling time, facility usage condition, and so on [1]. With the rapid development of EV technologies, the proportion of EVs is growing annually [2], and Figure 1 shows the worldwide EV sales statistics. However, limited endurance and charging stations and a much longer charging time compared with the short refueling time of petrol cars cause serious mileage anxiety for EV drivers [3]. As one of the most-significant infrastructures of the EV system, EV charging stations have attracted more attention recently. Some studies have shown that EV charging behavior has obvious periodicity [4], thereby an accurate EV charging station usage condition forecasting system can effectively alleviate range anxiety and improve road efficiency [5].

Benefiting from the huge number of smart sensors, real-time station-level monitoring has been realized [6]. Most canonical facility usage condition prediction methods are dependent on past traffic features to make predictions. However, EV charging station availability is much more complex than other time series forecasting issues because the future availability not only depends on the historical values, but is also influenced by the topological relationship and complex external influences [7]. For example, within the campus or Central Business District (CBD) road section, the usage of the charging station will be highly affected by the commute time. An obvious rise of availability can be observed around the off-duty time, which is the reverse inside the residential area, even though the two road structures are similar [8]. Another example is that bad weather, such as heavy rain, can increaseand delay people’s commute and further affect charging station usage [9]. It is quite a challenge to take into consideration the randomness caused by these external factors [10].

With the development of deep learning technologies, several forecasting methods have been proposed to solve this issue [11], such as the Auto-Regressive and Integrated Moving Average (ARIMA) method [12], the Convolutional Neural Network (CNN) method, the Long Short-Term Memory (LSTM) method, the GCN method [13], and the Transformer-based method [14]. Each algorithm has its own strengths and limitations. However, most of the models do not have the capability to obtain the augmented attributes during the data processing. Correspondingly, the perception of external factors is poor. In the next section, a detailed introduction to the related work is given.

Compared to the recent related works, we built a novel neural network extracting both spatiotemporal information and external influences to predict the charging station usage condition. The contributions can be summarized as follows:As far as we know, our study is one of the few research works on the deep learning approaches for the EV charging station availability forecasting problem.The AST-GIN’s structure is firstly proposed to deal with the EV charging station availability forecasting problem by combining the Attribute Augmentation Unit (A2Unit), the GCN, and the Informer network.The proposed AST-GIN model was verified and tested on real-world data. The comparison results showed that the AST-GIN has better prediction capability over different horizons and metrics.

The rest of the work is arranged as follows: The second section describes the related research on deep learning approaches for traffic facility usage forecasting and external factors’ influence during the time series prediction. The third section illustrates the problem statement and proposed model structure. The fourth section shows the detailed experiments with an analysis. The final section summarizes the contributions and possible future plans.

## 2. Related Research

### 2.1. EV Charging Issue

Recent research has shown that charging is a challenge for the operation of a fleet of EVs, since frequent charging sessions are needed [15]. Alleviating charging station congestion has become significant to improve the efficiency of charging infrastructure management [16,17]. Two main research directions for the EV charging problem have been studied recently. One direction focuses on modeling individual EV charging loads and charging stations. The objective is to predict the parameters of the charging load profiles for a smart charging management system [18]. Existing studies mainly apply statistical models [19], such as Gaussian mixture models [18], and deep learning approaches [20], such as a hybrid LSTM neural network [21,22], to forecast charging loads at EV charging stations. In [23], the authors reviewed the most-popular techniques for EV load modeling, including deterministic and probabilistic methods. From short-term to long-term perspective, researchers have proposed several forecasting methods. Utilizing the advantages of the Internet of Things (IoT) technology, the real-time interactional view of charging stations and the server-based forecasting application have been realized [24]. In [25], the authors proposed a daily joint adversarial generation interval-forecasting method for EV charging load distribution by considering the influence of the spatial correlation and characterizing the randomness. Some researchers even presented a mid- and long-term systematic method to predict the additional loads of EV charging by considering the EV charging profiles and future EV ownership [26]. The other direction analyzes modeling and predicting the charging occupancy profile at the chargers, which is quite similar to the parking availability prediction problem [27,28]. The purpose is to design the scheduling algorithm to allocate EVs among eligible chargers to realize the global or local optimal charging waiting plan [29,30].

### 2.2. Canonical Forecasting Model

For the traffic forecasting issue, the approaches have undergone several stages, and the methods can generally be divided into two types: canonical models and deep-learning-based models [31]. Canonical forecasting models usually build mathematical models and treat traffic behavior as the conditional process. There are many famous models, such as the Historical Average (HA) model, the K-nearest neighbor model, the ARIMA model, and Support Vector Regression (SVR) model [32]. Most of them consider the trend of the data and make the strong assumption that time series data are stable, which makes it difficult for them to respond to the rapid change of the inputs [33].

### 2.3. Deep Learning Forecasting Model

Recently, deep-learning-based forecasting methods have been widely applied to solve time series prediction problems [34]. Benefiting from their capability to extract nonlinear relationships across an input sequence, the Recurrent Neural Network (RNN) model, the Stacked Autoencoding Neural Network (SAE), the Gated Recurrent Unit (GRU) [35], LSTM [36], Transformer [37], and their variants have been verified to be much more efficient at extracting temporal information than canonical forecasting models. To adaptively predict comprehensive traffic conditions, some works have been performed to improve the results, such as integrating a GCN to extract the spatial dependencies [13,38].

### 2.4. External Factors in Forecasting

As mentioned above, external factors have an influence on the future usage conditions of EV stations. To integrate the information of a variety of external inputs, such as surrounding POIs [39] and weather conditions [40], previous studies have demonstrated great efforts, leveraging multi-source data to specifically design the model structure. In [41], the authors proposed an LSTM-based structure integrating an encoder to aggregate external information and treat multi-source data as the sequential inputs. In [35], the authors applied the feature fusion technology to process the input weather data for traffic prediction.

In conclusion, existing methods can be further improved by considering external information’s influence. Therefore, motivated by the related works and the challenges, the AST-GIN network for EV charging station availability forecasting, which integrates both spatiotemporal and external factors as the input to enhance the model’s perception capability during predicting, is proposed. In the next section, the architecture and principles of the proposed model are illustrated.

## 3. Methodology

### 3.1. Definition of EV Charging Station Availability

In this section, we first give the mathematical definition of EV charging station availability. Availability represents the occupancy status of EV charging facilities, such as charging connectors. If all connectors in a charging station are occupied for charging, the availability is regarded as 0. On the other hand, if all connectors are in the unused status, the availability is regarded as 1.

Based on the definition, the availability of the charging station can be calculated as:(1)$$x_{it}=1-\frac{M_{i,usedconnector}}{M_{i,allconnector}}$$
where $M_{i,usedconnector}$ is the number of charging connectors being used at station *i* at time *t*; $M_{i,allconnector}$ is the total number of charging connectors at station *i* at time *t*. Herein, the ultimate target variable is x, where $x_{it}$ is the availability of the charging station *i* at time *t*. Note that the value of $x_{it}\in \left[0,1\right]$.

Before introducing the model for predicting the availability of the EV charging stations, we subsequently describe the attributes that exert an impact on the our target variable x, i.e., the charging station availability.

### 3.2. Incorporating the Attributes

#### 3.2.1. Weather Condition Attribute

The purpose of this study was to predict future EV stations’ usage condition based on historical states and associated information. Based on the prior knowledge introduced, the demand of EVs has a strong periodicity, and external factors have a high correlation with the usage of EVs. As shown in Figure 2, the weather is classified into 5 types: sunny, cloudy, foggy, light rain, and heavy rain, which are labeled 1 to 5. To better present the relationship, the weather data were normalized in the range of 0 to 1. Therefore, the Y-axis for the availability data refers to the availability of the chargers in Figure 3. At the same time, the Y-axis for the weather data refers to the different kinds of weather. The higher the value, the worse the weather. The graph shows a general pattern by which the availability becomes higher when the weather becomes better and vice versa. Meanwhile, the time index in the figure refers to the recording time. In this work, the weather factor is organized as a matrix *W*, where $W_{it}\in \left[0,1\right]$ is the weather condition of location *i* at time *t*.

#### 3.2.2. Road Network and POI Attributes

In this paper, the road network was treated as a weighted undirected graph G={V,E}. EV stations work as nodes inside the graph, which are denoted by |V|=N, where *N* is the number of stations and *E* represents the graph edge set representing stations’ connectivity. The corresponding adjacency matrix *A*∈$R^{N\times N}$ can be constructed based on node and edge information. With the road map, based on the latitude and longitude of the charging points, the road distance between EV stations can be estimated. The adjacency matrix elements are calculated using the Gaussian kernel weighting function [42]:(2)$$A_{ab}=\left\{\begin{array}{cc} exp\left(-\frac{dist{\left(v_{a},v_{b}\right)}^{2}}{\sigma^{2}}\right),\;dist\left(v_{a},v_{b}\right)\leq \kappa & \\ 0,\;otherwise \end{array}\right.$$
where $dist(v_{a},v_{b})$ represents the distance between station $v_{a}$ and station $v_{b}$; σ is the standard deviation of $dist(v_{a},v_{b})$; κ is the filter removing small distances.

**Figure 2 sensors-23-01975-f002:**
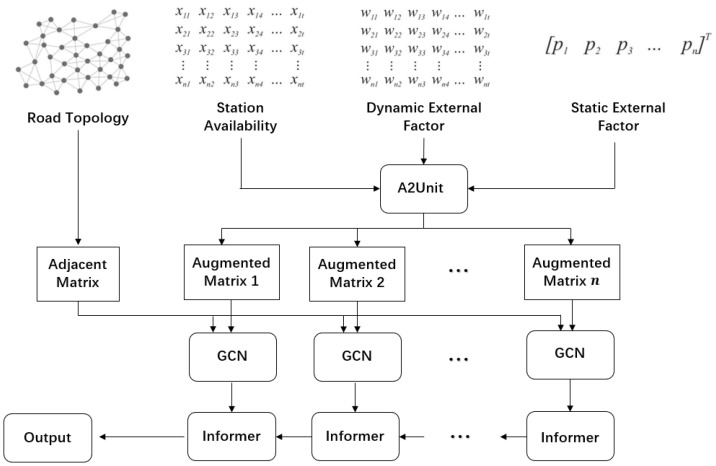
AST-GIN. architecture.

**Figure 3 sensors-23-01975-f003:**
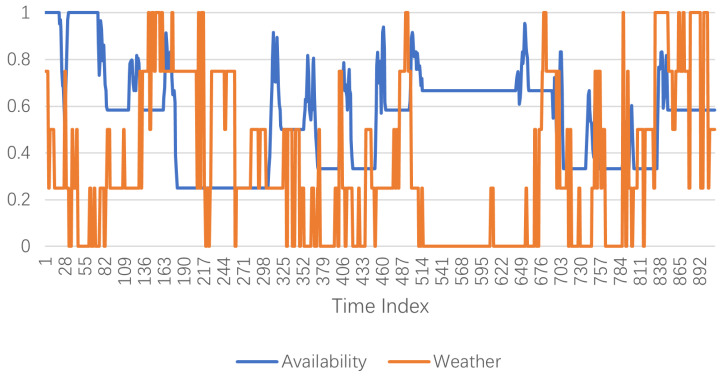
Weather and EV charging station availability.

Besides, we also integrated the POIs’ distribution information as an external factor, which is denoted as
(3)$$\left[{poi}_{1},{poi}_{2},\cdots ,{poi}_{n}\right]$$
where ${poi}_{i}$ is the POI category score for location *i*. Assume we have *k* POI categories, then we have ${poi}_{i}\in \left\{1,2,\cdots ,k\right\}$.

### 3.3. Problem Formulation

At time *t*, the EV charging station availability matrix, $X_{t}$∈$R^{N\times C}$, contains the high-dimensional information of EV station availability, where *C* is the hyperparameter, which manually defined. Thus, the known *L* steps’ historical usage data are defined as $X=\left[X_{t-L},X_{t-L+1},...,X_{t}\right]$ and used as partial inputs to predict the next *M* steps’ states $\left[{\hat{Y}}_{t+1},...,{\hat{Y}}_{t+M}\right]$.

Further, the influence of external factors is regarded as the affiliated attributes matrix *F*. These factors construct an attribute matrix $\left[F_{1},F_{2},...,F_{l}\right]$, where *l* is the dimension of the attribute information. At time *t*, the set of *j*-th affiliated information is $F_{j}=\left[j_{t-L},j_{t-L+1},...,j_{t}\right]$.

In conclusion, the issue of EV charging station availability forecasting considering external factors is refined to finding the relationship function *f* based on the historical usage data *X*, attribute matrix *F*, and road graphic structure *G*, to achieve the future usage values $\hat{Y}$:(4)$$\left[{\hat{Y}}_{t+1},...,{\hat{Y}}_{t+M}\right]=f\left(G,X,F\right)$$
where ${\hat{Y}}_{t+m}$ means the estimated value of $X_{t+m}$ at the future time t+M, m=1,2,…,M, and *M* is the prediction horizon.

### 3.4. AST-GIN Architecture

In this subsection, the principle of the AST-GIN model is introduced in detail. The AST-GIN model contains three layers: A2Unit, which can integrate the external information, the GCN layer, and the Informer layer. The historical time series data and external data are firstly fed into the A2Unit for attribute augmentation. Then, the processed information is fed into the GCN layer for spatial information extraction. Finally, the Informer layer will take the outputs from the GCN layer to extract the temporal dependencies. The overview of the architecture of the AST-GIN is illustrated in Figure 4.

#### 3.4.1. A2Unit

As mentioned, both historical data and external factors affect the EV charging conditions. Thus, different from traditional time series deep learning forecasting model, additional structure aggregating external factors are needed [41]. To comprehensively take external factors’ influence into consideration, dynamic attributes and static attributes are selected, respectively, for the objective region. EV stations’ historical availability tensor *X*, road structure *G*, and two types of attribute matrices are integrated into the A2Unit for augmentation.

We use α∈$R^{N\times p}$ to represent the static attribute matrix containing *p* categories’ attributes, and α is time-invariant. Similarly, β∈$R^{N\times (w*t)}$ represents *w* different dynamic attributes with cumulative effects, which change over time. To aggregate the cumulative influence of dynamic attributes, the *L* length historical window is selected. Thus, the final augmented matrix processed by A2Unit at time *t* is stated as:(5)$$E^{t}=\left[X_{t},\alpha ,\beta^{t-L},\beta^{t-L+1},...,\beta^{t}\right]$$
where $E^{t}\in R^{N\times (p+1+w*(L+1))}$, and the same processing procedure is applied for every time stamp inside traffic feature matrix *X*.

#### 3.4.2. GCN Layer

The distance between the vehicle location and the target charging station obviously influences the decision of the drivers [43]. To enhance the understanding of the EV charging behavior pattern, the spatial dependencies among charging stations were taken into consideration. Some related works have proposed CNN-based neural networks to deal with the spatial prediction issue [44,45]. However, the distribution of EV charging stations connected by the non-Euclidean road network cannot be processed well by a CNN. Thus, here, the GCN was selected to extract the spatial dependencies of the input data, which still retains the convolutional functionality [46]. The framework of the GCN is shown in Figure 5.

In principle, graph convolution neural network perform the convolution over the nodes of the graph to capture the spatial information, which is similar to image processing by convolution neural networks.

The general convolution theorem in the spatial domain states that
(6)$$f*g=F^{-1}\left\{F\left\{f\right\}\cdot F\left\{g\right\}\right\}$$
where *g* is the kernel operated on function *f*.

When performing the convolution in the spectral domain, the graph convolution formula can be expressed as
(7)$$g_{\theta }*x=Ug_{\theta }U^{T}x=Ug_{\theta^{\prime }}\left(\Lambda \right)U^{T}x$$
where $g_{\theta }$ is the graph convolutional kernel, *U* is the eigenvector matrix, and Λ is the diagonal matrix of eigenvalues.

To simplify the computation, people usually perform the first-order Chebyshev approximation [46]. The graph convolution formula now is reorganized as
(8)$$g_{\theta }*x=\theta \left(I_{n}+D^{-\frac{1}{2}}AD^{-\frac{1}{2}}\right)x=\theta \left({\tilde{D}}^{-\frac{1}{2}}\tilde{A}{\tilde{D}}^{-\frac{1}{2}}\right)x$$
where *D* is the diagonal degree matrix, $D_{ii}$ = $\sum_{j}$$A_{ij}$; $\tilde{A}$ = A + $I_{n}$; ${\tilde{D}}_{ii}$ = $\sum_{j}$${\tilde{A}}_{ij}$.

Furthermore, the GCN convolutional formula can be rewritten as:(9)$$H^{l+1}=\gamma \left({\tilde{D}}^{-\frac{1}{2}}\tilde{A}{\tilde{D}}^{-\frac{1}{2}}H^{l}W^{l}\right)$$
where γ is the activation function; $H^{l+1}$ is the *l*-th layer output.

#### 3.4.3. Informer Layer

It is significant to obtain the global temporal dependency while forecasting. With the rapid development of Transformer-like neural networks, which employ an encoder–decoder architecture, the time series prediction ability has been improved greatly based on the attention mechanism compared to some canonical deep learning methods [47], such as the GRU and LSTM. Thus, one of the latest variants of Transformer, Informer [48], was applied here as the temporal extraction layer to understand the global sequence. The structure of Informer is shown as Figure 6.

#### 3.4.4. Loss Function

During the model training, the objective was to eliminate the gap between the ground truth and predicted value. Thus, the loss function can be written as
(10)$$loss=∥Y_{t}-\hat{Y_{t}}∥+\lambda L_{reg}$$

$Y_{t}$ and $\hat{Y_{t}}$ represent the recorded value and the predicted value, respectively. $\lambda L_{reg}$ [49] is the L2 regularization term to avoid overfitting during training.

## 4. Empirical Analysis

To evaluate the AST-GIN model’s performance, the necessary experiments were performed on the EV charging station availability dataset. We chose five efficient time series forecasting baseline models for comparison. During the experiment, the performance of the AST-GIN model with a static external factor only, the model with a dynamic external factor only, and the model with both static and dynamic factors were evaluated separately.

### 4.1. Dataset and Preprocessing

#### 4.1.1. EV Charging Station Data

Dundee EV charging dataset: This dataset is a record of the EV charging behaviors in Dundee, Scotland, available at https://data.dundeecity.gov.uk/dataset/ev-charging-data (accessed on 10 October 2022). There are 57 charging points in Dundee, which can be divided into three types: slow chargers, fast chargers, and rapid chargers. In total, three valid datasets recorded during three different time periods, 01/09/17 to 01/12/17, 02/12/17 to 02/03/18, and 05/03/18 to 05/06/18, are accessible. Meanwhile, the geographical locations of all the charging points are also provided, and they are shown in Figure 7.

In the present study, the dataset we used was recorded during 05/03/18 to 05/06/18. There were 16773 charging sessions recorded in total, and each of the charging session records contains the charging point ID, charging connector ID, starting and ending charging time, total consumed electric power, geographical location, and type of charging point. There were 40 slow chargers with 5894 charging sessions recorded, 8 fast chargers with 1416 charging sessions recorded, and 9 rapid chargers with 9463 charging sessions recorded.

Moreover, Dundee’s weather data are available, which is the weather record in Dundee City. The weather is recorded every hour, and each record includes the general description of the weather, temperature, wind, humidity, and barometric pressure.

#### 4.1.2. Static External Factors

We classified the surroundings of the charging points into eight types: transportation services, catering services, shopping services, education services, accommodation, medical services, living services, and other. The category of the surroundings with the largest proportion was labeled as the POI value of a charging point based on its geographical location.

#### 4.1.3. Dynamic External Factors

The weather for Dundee was divided into five types: sunny, cloudy, foggy, light rain, and heavy rain, with different labels from 1 to 5. Since the time interval of the weather data from the source dataset was 1 h, the weather in the covered period was regarded as the same, which means that, if the weather at 17:50 is recorded as sunny, the weather at 18:20 would be labeled as sunny.

### 4.2. Settings

#### 4.2.1. Evaluation Metrics

In this study, five commonly used metrics were selected to evaluate the model’s forecasting performance, including the Root-Mean-Squared Error (RMSE), the Mean Absolute Error (MAE), the Accuracy, the Coefficient of Determination ($R^{2}$), and the Explained Variance Score (EVS). Each of the metrics is defined as follows:(11)$$RMSE=\sqrt{\frac{1}{MN}\sum_{j=1}^{M}\sum_{i=1}^{N}{\left(y_{ij}-{\hat{y}}_{ij}\right)}^{2}}$$
(12)$$MAE=\frac{1}{MN}\sum_{j=1}^{M}\sum_{i=1}^{N}\left|y_{ij}-{\hat{y}}_{i,}\right|$$
(13)$$R^{2}=1-\frac{\sum_{j=1}^{M}\sum_{i=1}^{N}{\left(y_{ij}-{\hat{y}}_{ij}\right)}^{2}}{\sum_{j=1}^{M}\sum_{i=1}^{N}{\left(y_{i}^{j}-\bar{Y}\right)}^{2}}$$
(14)$$Accuracy=1-\frac{{∥Y-\hat{Y}∥}_{2}}{{∥Y∥}_{2}}$$
(15)$$EVS=1-\frac{Var\{Y-\hat{Y}\}}{Var\{Y\}}$$
where $y_{ij}$ and ${\hat{y}}_{ij}$ represent the ground truth availability and the predicted one for the i−th charging station at time *j*. *M* is the time instant number; *N* is the charging station number; Y and $\hat{Y}$ are the set of $y_{ij}$ and ${\hat{y}}_{ij}$, respectively; $\bar{Y}$ is the average of Y.

#### 4.2.2. Baseline Settings

As far as we know, there is no directly published model for this specific problem; hence, typical models were selected for comparison. Comparing the proposed model with the typical sequence models of the GRU, LSTM, Transformer, and Informer involved in our model, we can testify to the importance of incorporating the spatial dependencies captured by the GCN. As expected, when we compared the proposed model with Informer, the significance of the GCN in the architecture can be verified. The baselines are described as follows:GRU: The commonly used time series model, which has been proven effective in traffic prediction problems and can alleviate the problem of gradient explosion and vanishing.LSTM: Together with the GRU, they are two popular variants of the RNN. LSTM has a more complex structure than the GRU.Transformer: The classic Transformer model with the self-attention mechanism [37].Informer: A new Transformer variant proposed to process the long-sequence prediction issue without spatial dependencies’ extraction.STTN: A new proposed framework utilizing two Transformer blocks to capture both spatial and long-range bidirectional temporal dependencies across multiple time steps [50].

#### 4.2.3. Hyperparameters

In this study, a three-layer GCN structure was used. For each Informer block, the number of encoder layers was 2, while the number of decoder layers is 3. During the experiment, the data were randomly divided into 50% for training, 33% for the evaluation, and the remaining 17% for testing. The network was optimized using the Adam optimizer. The dropout rate was set to 0.05. The learning rate started from $e^{-4}$ and decayed by 10 times every two epochs. The total number of epochs was 50 with an early stopping criterion. The batch size was chosen as 32, and the whole network was trained on a GPU RTX3060.

### 4.3. Experimental Results

We used five state-of-the-art baselines to compare the performance of our proposed AST-GIN model, including the GRU, LSTM, Transformer, Informer, and STTN. Based on the 30-minute time interval of the EV charging availability dataset, we deployed the selected models to predict the availability in the next 30 min, 60 min, 90 min, and 120 min horizons. The numerical results are shown in Table 1.

For both short-term (30 and 60 min) and long-term (90 and 120 min) EV charging station availability forecasting, the AST-GIN model effectively captured the temporal dependance of the data and outperformed the baseline models, as measured by all the metrics. In the 30 min forecasting, AST-GIN model with the dynamic external factor achieved an accuracy of 0.8388, while the best model, the STTN, among the baseline models achieved an accuracy of 0.7521. The AST-GIN outperformed the STTN in the 90 min horizon with a 11.54% higher accuracy. In the long-term forecasting, 120 min for example, the AST-GIN still had the best performance with the highest accuracy compared to the other baseline models. The accuracy of the AST-GIN, 0.7517, was 8.97% higher than the best model, the STTN, among the baseline models.

Among the three kinds of external factors used, which were the static factors only, dynamic factors only, and the combination of both of the factors, the use of the external factors’ combination led to a better performance in general for the prediction horizons of 30 min and 60 min. The combination of both static and dynamic factors led to higher prediction accuracy in the relatively short-term horizon. The difference among static the factors only, dynamic factors only, and the combination became negligible in the relatively long-term horizon.

For a better comparison, we plot the prediction accuracy of all models in terms of all prediction horizons in Figure 7. Measured by the explicitly interpretable accuracy metric, it can be observed that the overall forecasting capability of the proposed model was better than the baselines over all prediction horizons. For the prediction horizons of 90 min and 120 min, the proposed AST-GIN performed better by incorporating either the POI feature or the weather feature than when incorporating both factors. The external factors’ combination should be considered extensively for short-term prediction. Such results showed that we could distinguish the attribute inputsin terms of short-term prediction or long-term prediction.

### 4.4. Results’ Analysis

To analyze the prediction results of the proposed model and the benchmarks, we plot the cumulative distribution function of the absolute prediction errors in Figure 8, where the absolute error was calculated as $|y_{ij}-{\hat{y}}_{ij}|$. Note that, as shown in Table 1, the prediction results of Informer were always better than that of Transformer; hence, in this part, we excluded the results of Transformer in Figure 8.

From the CDF of the errors, we can find that the absolute error of the proposed AST-GIN model occurred earlier and faster than the three baselines when the prediction horizons were 90 min and 120 min. This means that the long-term prediction performance is of great significance for the EV charging station availability estimation, which is because the drivers can know in advance with an adequate time to head to the available station.

In addition, inevitably, noise during the data collection, such as sensor error and system delay, may undermine the model’s forecasting capability. To verify the noise immunity of the AST-GIN model, the model’s robustness was tested via perturbation experiments. The normalized random noise, which obeys a Gaussian distribution, was added to the data to check the robustness. The resultant fluctuation was small.

## 5. Conclusions

In this paper, a deep learning model, the AST-GIN, was proposed and verified for EV charging station availability forecasting considering external factors’ influence. The model contained an A2Unit layer, GCN layer, and Informer layer to augment time series traffic features and extract the spatiotemporal dependencies of EV charging station usage conditions. The AST-GIN and the baselines were tested on the data collected in Dundee City. The experiments showed that the AST-GIN has better forecasting capability over various horizons and metrics. To summarize, the AST-GIN can effectively comprehensively consider the external attribute influences and predict the EV charging station usage condition. Future plans regarding the robustness are ongoing to further improve the prediction system.

## Figures and Tables

**Figure 1 sensors-23-01975-f001:**
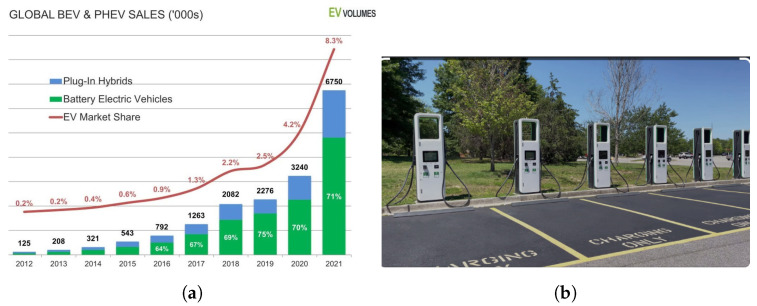
EV sales statistics and EV charging station example. Global EV sales increased 108% from a 4.2% market share in 2020 to a 8.3% market share in 2021 [2]. (**a**) Global EV sales; (**b**) electric vehicle charging hubs open in Dundee.

**Figure 4 sensors-23-01975-f004:**
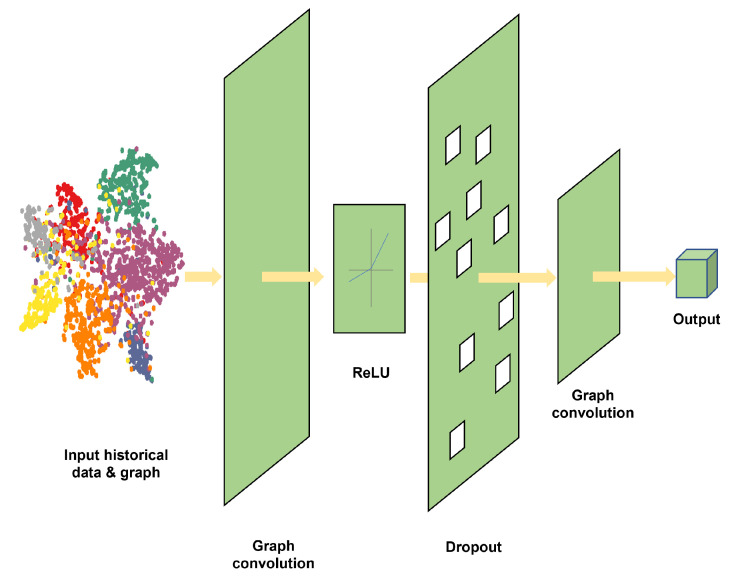
GCN layer architecture.

**Figure 5 sensors-23-01975-f005:**
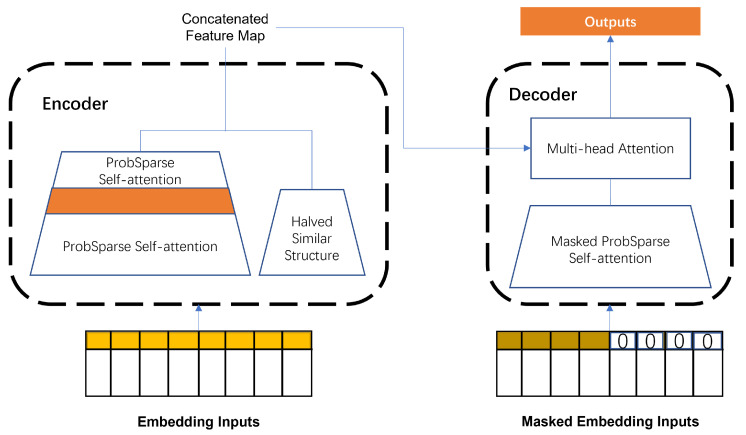
Temporal Informer layer structure.

**Figure 6 sensors-23-01975-f006:**
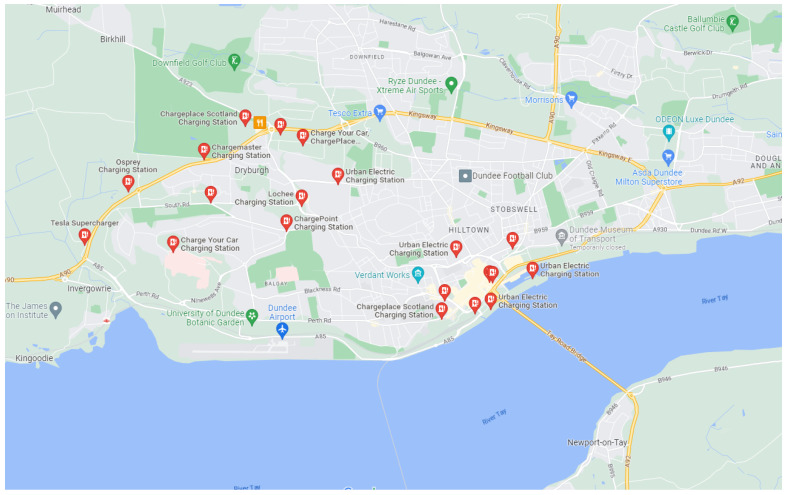
Locations of EV charging points. Charging points distribution in Dundee City is shown in the figure.

**Figure 7 sensors-23-01975-f007:**
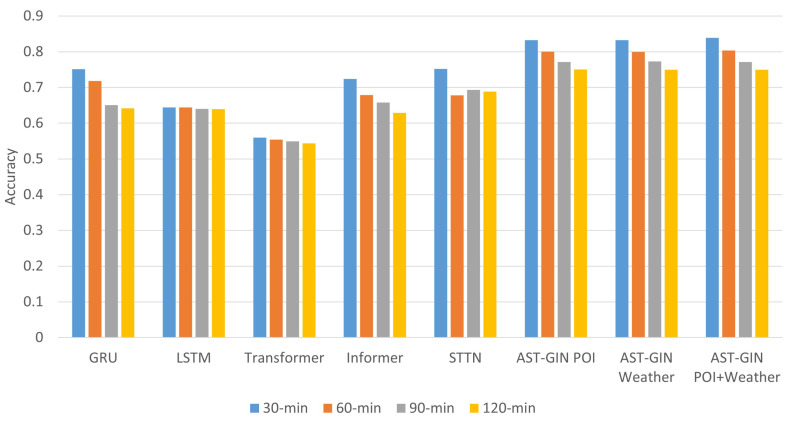
Accuracy statistics of all models.

**Figure 8 sensors-23-01975-f008:**
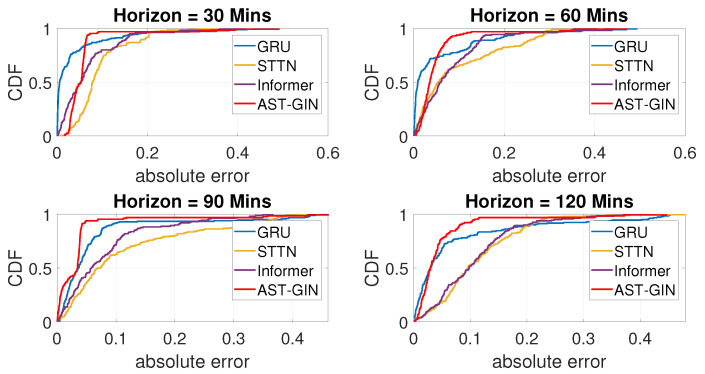
Cumulative distribution function of the absolute error by each model.

**Table 1 sensors-23-01975-t001:** Forecasting results of the AST-GIN and baseline models.

Horizon (min)	Metric	GRU	LSTM	Transformer	Informer	STTN	AST-GIN		
							POI	Weather	POI + Weather
30	RMSE	0.1726	0.2431	0.2923	0.2112	0.171	0.1215	0.1224	**0.1174**
	$R^{2}$	0.7665	0.4918	0.4021	0.6583	0.7183	0.8778	0.8709	**0.8803**
	EVS	0.7579	0.4879	0.3746	0.6511	0.7175	0.8787	0.8704	**0.8801**
	MAE	0.1041	0.1683	0.2365	0.1556	0.1331	0.0784	0.0759	**0.067**
	Accuracy	0.7531	0.6493	0.5589	0.7293	0.7521	0.8382	0.8322	**0.8388**
60	RMSE	0.1820	0.2321	0.2862	0.2326	0.2221	0.1446	0.1471	**0.1438**
	$R^{2}$	0.6851	0.5047	0.3952	0.5467	0.6248	0.8149	0.8174	**0.8227**
	EVS	0.6789	0.4941	0.3782	0.5376	0.6276	0.8149	0.8174	**0.8225**
	MAE	0.1168	0.1735	0.2385	0.1870	0.1679	0.0827	0.0864	**0.0757**
	Accuracy	0.7138	0.6424	0.5534	0.6798	0.6728	0.8020	0.7994	**0.8037**
90	RMSE	0.2269	0.2336	0.2848	0.2613	0.2118	0.1682	**0.1674**	0.1687
	$R^{2}$	0.5362	0.496	0.3335	0.4806	0.5718	0.7652	**0.7653**	0.7605
	EVS	0.5085	0.485	0.3662	0.4695	0.5634	0.7641	**0.7652**	0.7604
	MAE	0.1548	0.1741	0.2377	0.1976	0.1683	**0.0957**	0.0982	0.1017
	Accuracy	0.6508	0.6406	0.5491	0.6581	0.693	0.7713	**0.7731**	0.7713
120	RMSE	0.2372	0.2354	0.2896	0.2882	0.3264	**0.1834**	0.1852	0.1851
	$R^{2}$	0.5114	0.4743	0.3237	0.4553	0.5581	**0.7162**	0.7138	0.7134
	EVS	0.4823	0.4675	0.3624	0.3934	0.5524	**0.7154**	0.7131	0.7131
	MAE	0.1565	0.1769	0.2369	0.2128	0.1643	0.1134	**0.1106**	0.1123
	Accuracy	0.6481	0.6329	0.5473	0.6238	0.6839	**0.7517**	0.7496	0.7496

## Data Availability

The dataset used in this paper can be found at https://data.dundeecity.gov.uk/dataset/ev-charging-data (accessed on 10 October 2022).

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
