# Peer review of "AST-GIN: Attribute-Augmented Spatiotemporal Graph Informer Network for Electric Vehicle Charging Station Availability Forecasting"

_sensors, 2023, doi:10.3390/s23041975_

Round 1

Reviewer 1 Report

In this paper, authors successfully propose a deep learning-based EV charging station availability forecasting algorithm based on traffic sensors data. Even though there are lots of time-series prediction methods recently, few works consider the comprehensive external influence on traffic parameters. The novelty and clarity have been clearly delivered in this paper. The design of model also considers the significance of spatial and temporal dependencies, which enhance the interpretation of forecasting capabilities. Additionally, the experiment is convincing with clear experiment settings and CDF analysis. In conclusion, I recommend accepting this submission with minor revision as shown below:

1. Dropout rate that is used in the GCN layer in Figure 4 should be elaborated in section “Hyperparameters”.

2. Some typos should be modified, such as the “Hyperprameters” in line 244 should be “Hyperparameters”, “fluence” in line 96 should be “influence”, etc.

Reviewer 2 Report

The authors have proposed a machine learning based approach for electric vehicles load demand forecasting. I have following comments for this paper.

1- As the authors know there is high correlations between the load demand parameters including the arrival time, departure time, and driving distance. However, I couldn't find the consideration of the correlation in the model.

2- Is the collected data enough for training of the network?

3- You have considered 50% of the data for training, 17% for evaluation and 13% for testing. What are your metrics for choosing these amount of percentages? 

4- Again, I couldn't find the contribution and novelty of this work clearly.

5- The literature survey is weak. Too many high quality papers have been published in the literature that you can enrich the literature. Please use the following and others to improve the literature.

- Huang, Nantian, Qingkui He, Jiajin Qi, Qiankun Hu, Rijun Wang, Guowei Cai, and Dazhi Yang. "Multinodes interval electric vehicle day-ahead charging load forecasting based on joint adversarial generation." International Journal of Electrical Power & Energy Systems 143 (2022): 108404.

- Zheng, Yanchong, Ziyun Shao, Yumeng Zhang, and Linni Jian. "A systematic methodology for mid-and-long term electric vehicle charging load forecasting: The case study of Shenzhen, China." Sustainable Cities and Society 56 (2020): 102084.

- Savari, George F., Vijayakumar Krishnasamy, Jagabar Sathik, Ziad M. Ali, and Shady HE Abdel Aleem. "Internet of Things based real-time electric vehicle load forecasting and charging station recommendation." ISA transactions 97 (2020): 431-447.

Ahmadian, Ali, Behnam Mohammadi-Ivatloo, and Ali Elkamel. "A review on plug-in electric vehicles: introduction, current status, and load modeling techniques." Journal of Modern Power Systems and Clean Energy 8, no. 3 (2020): 412-425.
